# Informal Competition Effect on SMEs' Innovation: Do Credit Constraints Matter? Evidence from Eastern European Countries

**Zaineb Hlioui [1,\*], Mohamed Gabsi [2] and Abdelwahed Omri [1]**

[1]   GEF2A-Lab, Higher Institute of Management of Tunis, University of Tunis, Le Bardo 2000, Tunisia
[2]   Higher Institut of Management, University of Tunis, Le Bardo 2000, Tunisia
\*   Correspondence: zainebhlioui@isg.u-tunis.tn; Tel.: +21-(65)-4948833

**Abstract:** This paper examines the influence of informal competition on SME innovation in the Eastern European transition economies. Using the BEEPS VI, which covers the period from 2018–2020, we investigated the conditional mediation of credit constraints moderated by business plan elaboration. Looking at SMEs' product innovation, process innovation, radical innovation, and green innovation, we find that informal competition's direct effect enhances all the innovation proxies. Besides, the informal sector increases SMEs' credit constraints, which indirectly leads to less corporate innovation. The negative indirect effect is alleviated by the business strategy development. Finally, using bootstrap resampling, we confirm the significant conditional mediation effect of credit constraints on the informal competition and the innovation proxies.

**Keywords:** innovation; informal competition; credit access; business plan; emerging countries

## 1. Introduction

The transition of the Eastern European countries has achieved great success during the past two decades. In 1989, after the collapse of the communist system, several tasks were required to generate growth. Spisak [1] indicated that East Europe is experiencing the fastest growth in the European region. Yet, their economic engine seems to be growing weary. The previously established strategies are no longer adequate to drive economic growth at the current stage of the transition. Since rising living standards could induce these economies to lose their competitiveness, they need now strategic innovative plans to skip the middle-class trap.

Given that SMEs are the backbone of the Eastern European economies, encouraging their innovativeness is a key factor for achieving the current transition stage successfully. Nevertheless, SMEs suffer from financial insufficiency. According to the EIB, IMF and EBRD report [2], the lack of finance is still one of the major obstacles for SMEs. The percentage of firms without credit access and banking relationships compared to the Western region is relatively high. Thus, funding innovation, which is considered a risky investment, could be harder. In the same line, Weill and Godlewski [3] underlined that the Central and Eastern European countries classify the lack of funding resources among the main three hurdles that impede SMEs' growth. Indeed, the most significant challenge is the existence of financial constraints [4]. SMEs are facing difficulties to obtain funding resources, especially for innovative risky projects [5]. Moreover, the socio-economic characteristics of the region such as the corruption level and market competitiveness are affecting corporate profitability and financial capacities, which in turn influence innovation. According to Nazarov and Obydenkova [6], democratic and competitive levels affect growth and are related to firm innovativeness in post-communist countries.

In this regard, competition from the informal sector is perceived as a real strategic issue that may influence the businesses' innovation strategies [7]. Informal competition is considered an illegal phenomenon through which unregistered firms escape from taxation

and state regulations [8]. Unfortunately, it is increasingly expanded in the transition economies [7] adding to other detrimental issues such as corruption, inadequate employee education, infrastructure, and access to finance [9]. The existence of informal competitors reduces the formal companies' market share and harms their profitability and repayment capacities. From another perspective, innovation can be considered a tool of differentiation. Since innovation cannot be easily imitated, formal firms can use it as a tool to overcome informal rivals. Thus, a positive direct effect of informal competition on innovation might exist. The EIB, IMF, and EBRD report [2] underlined that informal competition is one of the top three main obstacles that hinder the environmental business and green innovation.

Given that informal competition damages the company's profitability and market share, which impedes access to new financial sources and therefore harms corporate innovation, we assess whether the financial constraints mediate the informal competition-innovation relationship. Yet, a positive direct link might exist. Innovation can be considered a diversification tool that helps to overcome informal rivals. Hence, investigating the possible mediation link between informal completion and innovation is of interest. Moreover, we consider the moderating effect of formalized business strategy on the credit constraints and innovation link. Cosci et al. [10] underlined that elaborating business strategy could alleviate credit constraints and foster innovation.

Our study is an attempt to understand whether formal firms in East Europe respond to the competition from unregistered firms by differentiating through innovation. In this paper, we examine a conditional indirect effect of informal competition on innovation from 2018 to 2020 (just before the pandemic outbreak). Indeed, in 2017 some Central and Eastern European leaders gathered in Warsaw, and underlined the importance of a replacement for the outdated growth model and asked how to establish the new model [1]. Hence, we select the post-gathering period in our study to help answer an urgent need in this context. We consider the different social, economic, and political characteristics of those countries. Moreover, we distinguish between the European Union (EU) members and the non-EU ones. In fact, the EU state seems to have a homogenous institutional context, contrary to the other Eastern European countries. We point out that the Western Balkans countries (WBC) are struggling against grand challenges that slow down their economic development and which prevent them from being part of the EU members [9,11].

In this paper, we present four types of corporate innovation. We use product innovation, process innovation, radical innovation, and green innovation. Indeed, radical innovation refers to innovation that is new to the company as well as to the market, riskier, and less likely to be imitated by the informal sector. Moreover, we consider green innovation. Despite the SMEs' important impact on the ecological issue, they are not considerably aware of this role. They are responsible for more than 60% of the overall businesses' environmental impact in Europe [12]. To our knowledge, there is no previous study that has examined this conditional indirect link in the Eastern European context. While studies about the separated links are well-developed in the literature there was no previous attempt to investigate the conditional indirect link.

Our results suggest that credit constraints have a conditional partial mediation effect on the informal competition-innovation link. Informal competition has a positive direct effect on the innovation categories. Nevertheless, it makes SMEs' credit conditions stricter and final resource availability tougher. Being credit-constrained in its turn hinders corporate innovation. Therefore, there is a negative indirect effect that reduces the positive direct effect. Finally, the business plan elaboration reduces the negative indirect effect.

The remainder of our paper will be structured as follows: in the second section, we explore the theoretical background and develop our assumptions. Then, in Section 3, we present our sample and data sources. In the fourth section, we provide a clear justification for the selected variables. Followed by our methodology and models. The descriptive statistics and main results are reported in Sections 4 and 5. Finally, we present a brief conclusion in the last section.

## 2. Literature Review

Previous studies consider informal economies as a relevant factor in the industries' competitive dynamics in transition economies [13]. Nevertheless, its influence on formal economies' strategic management remains under-investigated [14]. Formal firms have to overcome this unfair competition to sustain themselves. In this regard, innovation might present an appropriate differentiation tool. Yet, given the riskiness of innovative projects and the considerable financial obstacles faced by SMEs, companies are struggling to surpass their informal competitors. To better understand the studied link, we consider four innovation proxies. Indeed, there are different innovation types with different specificities. Hence, the informal competition could influence them differently. In our study, we focus on innovation outputs. More specifically, we consider product innovation, process innovation, radical innovation, and green innovation. The selection of these innovation types depends on the risk level and framework pressure under the current ecological crises.

Prior studies, which examine the direct effect of informal competition on innovation did not provide a firm conclusion. For instance, the effect of informal completion on product innovation, which refers to the introduction of a new product, was examined by McCann and Bahl [8]. They stated that informal competition positively influences new product development. In fact, the positive relationship underlines that firms respond through product innovation to the informal competition, which presents a threat to businesses in the formal sector. In the same line, Mendi and Costamagna [15] underlined that formal firms' process and product innovation decisions are strongly directed and influenced by the existence of informal firms. We point out that consistent with Mendi and Costamagna [15] process innovation is defined as the introduction of a new process or the significant enhancement of an existing one. Yet, according to the EIB, IMF, and EBRD report [2], SMEs in East Europe are suffering from the informal sector, political instability, as well as the inadequately educated work force, which hinder their business development and innovation. The same report underlined that global value chains are contributing to define the growth and innovation model.

Other than product and process innovation, we consider radical innovation, defined as the product or process innovation that is not only new to the company but also to the market. This innovation type is riskier, since the company is not certain about the market acceptance of the innovation. Cuervo-Cazurra [16] underlined that companies use innovation, especially radical innovation, to counter informality.

Regarding green innovation, Li et al. [17] presented a literature review on green innovation. They indicated that green innovation is influenced by external factors such as policy intervention and market competitiveness as well as internal factors such as knowledge and awareness. In the same perspective, Guinot et al. [18] provided a theoretical framework for green innovation and underlined the research gap regarding green innovation explanatory factors. In our study, we define green innovation as the development of new measures for energy efficiency by the company itself.

The previous investigation focused on financial constraints as one of the main reasons that impedes SMEs' innovation. Molodchik [19] highlighted the negative effect of financial constraints on source knowledge availability. Along the same vine, Kahupi et al. [20] presented evidence to support the environmental engagement integration in the corporate business plan. They proved that the idea of sustainability is not enough to provide a competitive advantage. Hence, it might not be an adequate tool for diversification in absence of a clear formalized business plan. Sharing the same perception, Zhang et al. [21] found that, even if the stakeholders value sustainability, consumers will not spend more on sustainable products. They recommended policy interventions to make sustainable products more competitive. Those policies should concern the business strategies development for optimal integration as well as consumer awareness. The limited awareness of the opportunities and benefits of green innovation is a major reason [22,23]. Machová et al. [24] and Chen et al. [25] supported this stream of literature. They found that consumers purchase sustainable products if the products are marketed successfully. Lenox and

Toffel [26] also demonstrate that as information about competitive sustainable practices and sustainable education are diffused across units of the multinational corporation, their usefulness increases.

Considering prior research, we investigate the effect of informal competition on the four innovation categories. We examine the presence of a conditional indirect effect based on credit constraint mediation and business strategies' second stage moderation.

### 2.1. The Informal Competition Effect on Innovation

A wide strand of recent literature draws attention to the link between competition from unregistered firms and innovation in formal sectors [7,8,27]. Nevertheless, their empirical studies have led to controversial results. On the one hand, Pérez et al. [7] pointed out that informal competition boosts innovation. Based on the RBV theory, they explained that informal competition is a driving force for firms in the formal sector. Indeed, innovation can provide a tool for differentiation that cannot be imitated. It is a unique resource. More recently, Miocevic et al. [28] indicated that when intellectual property rights protection is weaker, informal competition improves product innovation. On the other hand, Mendi and Costamagna [15] suggested that informal competition reduces the firms' new product or process creation. The formal businesses are less likely to opt for differentiation strategies to maintain a competitive advantage and avoid their rivals' imitations. According to the dynamic capability theory, differentiation through innovation depends on inherent replicability as well as intellectual property rights' protection. Therefore, using innovation to become distinguished might not be appropriate, especially for financially constrained firms, depending on the context characteristics. The unregistered companies' competitive behavior is "unhealthy" since it threatens medium-sized businesses' activities and reduces their ability to create new products or develop new processes. Furthermore, considering Zhang et al.'s [21] findings, if consumers consider sustainable products' purchase optional due to the lack of environmental awareness, diversification through green innovation will not help overcome the informal competitors. It could even create a competitive disadvantage due to green innovation expenses.

Moreover, Kresic et al. [11] classified the competition from unregistered companies at the top of the obstacles to businesses' activities. They reported that the informal sector threatens the businesses' activities and induces higher costs for innovative firms. Given that, innovation types could respond differently to informal competition. We base our first assumption on the RBV theory:

**Hypothesis H1 (H1).** *Informal competition has a positive significant direct effect on firms' innovation.*

**Hypothesis H1a (H1a).** *Informal competition has a significant direct effect on product innovation.*

**Hypothesis H1b (H1b).** *Informal competition has a significant direct effect on process innovation.*

**Hypothesis H1c (H1c).** *Informal competition has a significant direct effect on radical innovation.*

**Hypothesis H1d (H1d).** *Informal competition has a significant direct effect on green innovation.*

According to the previous literature, the negative relationship between competition from unregistered firms and innovation may be driven by the existence of financial constraints. The profit decrease generated by this illegal practice may induce lower innovativeness due to the funds' insufficiency. Hence, we investigate the effect of informal competition on financial constraints.

### 2.2. Informal Competition as a Determinant of Financial Barriers

Financial constraints are considered one of the main obstacles to businesses' prosperity [10]. They refer to the insufficiency or the lack of existing funds and the requirement of external financing resources that might not be easily provided [29]. According to Popov [30],

the funding sources might not be accessible especially for SMEs due to the high collateral requirements, the complexity of procedures, or the fear of being rejected by financial institutions. This threat is more pronounced when firms operate under competitive pressure from unregistered companies. Indeed, SMEs assume extra financial costs when they compete against informal firms, which affect their repayment capacity and financial stability [10]. Distinguish et al. [27] stated that informal competition creates a barrier to the firms' credit access since the financial institutions consider these companies as doubtful borrowers. Hence, high-interest rates are applied for firms that operate under competitive pressures due to the considerable risk premium. In addition, more collaterals are required since lenders cannot check their creditworthiness [31]. Hence, we assume the negative effect of credit constraints on credit approvals.

**Hypothesis H2 (H2).** *Formal firms that are against unregistered competitors are more likely to face credit constraints.*

### 2.3. Financial Constraints and Innovation Nexus

Firms need to use a variety of resources to promote their innovative activities. However, innovation is characterized by riskiness and uncertainty. Hence, the introduction of new products or processes faces stricter financial frictions. Wherefore, a large body of research studied the financial constraints' influence on innovation investments and when they matter [5,32–35]. Lee et al. [33] found that innovative small businesses sought external funds during the post-crisis period, especially from banks, since they faced higher financial hurdles. Yet, Gama et al. [36] pointed out that firms are reluctant to apply for loans in the Eastern European region. They demonstrated that firms might refuse to apply for a loan because of difficult procedures and costly requirements. In the same line, Brown et al. [4] brought insight into the explanatory factors of the banks' credit reluctance in the European regions. Indeed, Eastern European firms may be discouraged to apply for credit due to the high interest rate or hard collateral requirements. Generally, these firms are SMEs or firms with high financial opacity. They also show that improvements in the credit-granting process and collaterals can mitigate financial constraints and reduce their negative effect on innovative projects. Diallo and Al-Titi [37] focused on credit access effects on technological innovation and suggested that bank competition affects credit availability and thereby corporate innovation. Savignac [38] mentioned that companies, which are exposed to higher credit constraints, are 20% less likely to innovate. Along the same line, Araujo et al. [39] showed that lenders adopt credit policies that create obstacles to technological innovation. Thus, collateral-poor firms will be unable to undertake innovative projects. Hence, internal funds should satisfy most of the credit-constrained companies' needs. According to Ahrends et al. [40], corporate cash holdings are important for long-term investment funding such as innovation. They play a major role in reducing the firm's need for external funds, especially during difficult periods to avoid high financial expenses and capital market frictions. Along the same stream, Brown et al. [4] stated the reason why cash holdings are necessary for R&D funding. They argued that these activities are known for their high volatility and adjustment cost so undertaking R&D projects becomes expensive for firms facing financial constraints. In contrast, businesses that are less exposed to financing pressures seem to rely less on cash holdings for R&D-smoothing. On the basis of the theoretical review related to the link between financial frictions and innovation, we assume that credit access is negatively linked with innovative activities.

**Hypothesis H3 (H3).** *Credit access have a negative effect on innovation.*

**Hypothesis H3a (H3a).** *Credit access have a negative effect on product innovation.*

**Hypothesis H3b (H3b).** *Credit access have a negative effect on process innovation.*

**Hypothesis H3c (H3c).** *Credit access have a negative effect on radical innovation.*

**Hypothesis H3d (H3d).** *Credit access have a negative effect on green innovation.*

Finally, we assume that the informal competition indirectly affects the firms' innovativeness through financial frictions since, as reported by the literature above, credit constraints might play the role of a partial mediator in the informal competition and be an innovation link. Competition from unregistered firms induces financial constraints, which may specifically influence the formal firms' innovation activities.

*2.4. The Conditional Mediation of Credit Access on the Informal Competition and Innovation Link*

According to the previous literature, the relationship between competition from unregistered firms and innovation may be driven by the existence of financial constraints. In this regard, Demirgüç-Kunt et al. [41] underlined that access to finance is crucial for businesses that need to seize novel opportunities and make investments in innovation in order to differentiate from their unregistered competitors and avoid their imitations. Along a similar line, Friesen and Wacker [42] highlighted the importance of access to finance for formal firms that aim at developing competitive reactions in the presence of informal competition. Moreover, they explained that financially constrained firms are the most pressured by informality and are less likely to respond to informal competition. Indeed, the lost market share due to the illegal competition may induce lower innovativeness and fund insufficiency. In addition, the business plan elaboration could considerably affect the credit constraints and innovation link. Hlioui and Yousfi [43], underlined the importance of business plan elaboration, especially for sustainable practices integration and responsible innovation. They indicated that formalized business plans should integrate sustainable strategies and that they could moderate the financial obstacles and innovation relationship. In the same line, Cosci [10] highlighted the importance of business plan elaboration and its effect on lenders' trust and evaluation, which affect innovation. Yet, there is no previous study that investigates the conditional mediation of credit constraints on the informal competition–innovation link. Hence, our fourth assumption:

**Hypothesis H4 (H4).** *The effect of informal competition on innovation is mediated by financial constraints and this indirect effect is moderated by the business strategy elaboration.*

**Hypothesis H4a (H4a).** *The effect of informal competition on product innovation is mediated by financial constraints and this indirect effect is moderated by the business strategy elaboration.*

**Hypothesis H4b (H4b).** *The effect of informal competition on process innovation is mediated by financial constraints and this indirect effect is moderated by the business strategy elaboration.*

**Hypothesis H4c (H4c).** *The effect of informal competition on radical innovation is mediated by financial constraints and this indirect effect is moderated by the business strategy elaboration.*

**Hypothesis H4d (H4d).** *The effect of informal competition on green innovation is mediated by financial constraints and this indirect effect is moderated by the business strategy elaboration.*

**3. Materials and Methods**

*3.1. Sample and Data*

Our data were sourced from the sixth wave of the Business Environment and Enterprise Performance Survey (BEEPS VI) (Please note that creating a free account is required to get access to the data: https://www.beeps-ebrd.com/data/ (accessed on 9 October 2022). This wave covers the period from 2018 to 2020 and presents an anonymous survey produced within the framework of the partnership between the European Bank for Reconstruction and Development (EBRD) and the World Bank. Its questionnaire provides various corporate-level measures of innovation and a wide range of information about illegal practices such as informal competition, bribery, and tax evasion based on the top managers' answers. It also contains several questions about the financial resources and



obstacles to credit access. In addition, to have a representative sample, this dataset gave special care to the adequate computation of weights aiming for accurate totals within each region, industry, and size stratum. These data were previously used [44–47].

The OECD report [48] underlined that Eastern European SME potential is underexploited. They account for 1/3 of value added and 1/2 of employment. Yet, their innovation-reporting is considerably limited compared to the large companies. Hence, this study focuses on SMEs. We excluded companies with more than 250 permanent employees. In addition, since it is hard to find published financial and/or extra financial statements for SMEs, survey data can provide information through direct questions. Therefore, the truthfulness of questions regarding opinions is required. BEEPS data provided a distinction between truthful answers and doubtful ones. Thereby, only reliable observations were maintained. We point out that primary industries such as mining and agriculture as well as public sectors such as education and health care were excluded from the data. Finally, companies that did not answer the financial and innovation modules were also excluded. Our sample covered 7795 observations from 22 post-communist Eastern European countries. Table 1 presents the sample distribution by industry following the International Standard Industrial Classification construction (ISIC).

**Table 1.** Sample composition by industry.

|  | Count | Frequency |
|---|---|---|
| Fabricated Metal Products | 432 | 5.54% |
| Food & Beverages | 722 | 9.26% |
| Garments | 232 | 2.98% |
| Hotels | 63 | 0.81% |
| Machinery & Equipment | 540 | 6.93% |
| Manufacturing | 1165 | 14.95% |
| Non-Metallic Mineral Products | 195 | 2.50% |
| Other Manufacturing | 683 | 8.76% |
| Other Services | 1921 | 24.64% |
| Retail | 1745 | 22.39% |
| Textiles | 97 | 1.24% |
| Total | 7795 | 100.00% |

*3.2. Variables Description*

3.2.1. Dependent Variable

Innovation: Scholars measured innovation using different proxies such as research and development spending, which are innovation inputs [32,49] or patents [50], and the product and process innovation [7,8] as technological innovation outputs. Other studies distinguished between different innovation outcomes, such as incremental innovation, when it is new to the firm but exists in the market, radical innovation when it is new to the market [51,52], and green innovation when this innovation is driven by environmental concerns [53]. Each innovation type has specified characteristics such as risk level and return on investment periods.

In our research and following Bocquet et al. [54] and Dalgıç and Fazlıoğlu [55], we considered product innovation if there is a new product development and process innovation if the company indicated that it has developed a new process. We defined these two categories based on two direct questions:

If the firm has introduced a new product or significantly enhanced an existing one over the last three fiscal years, then it has product innovation. For the second question, if the company has introduced a new process or significantly enhanced an existing one, then it has process innovation.

Hence, we introduced two dummy variables. Each variable takes "1" if the answer to its related question is yes and "0" otherwise. While the product innovation "PROD_INNO" refers to a change, the product or the introduction of new features can lead to an improve-

ment in its performance. The process innovation "PROC_INNO" refers to the process of production, such as modifications in the value chain activities inbound logistics or manufacturing process. This variable selection is in line with the OECD definition.

> *"A new or improved product or process (or combination thereof) that differs significantly from the unit's previous products or processes and that has been made available to potential users (product) or brought into use by the unit (process)."*

Moreover, we considered two other innovation proxies: radical innovation "RAD_INNO" and green innovation "GREEN_INNO". Indeed, radical innovation refers to the implementation of a product or process innovation that is not only new to the company but also to the market. This kind of innovation is considered riskier since the market uncertainty is higher. Using a third question, we verifed whether the new product or process introduced by the company is also new to their market or not. Thus, the third innovation proxy is radical innovation. A dummy variable that takes "1" if the answer is yes and "0" otherwise. This variable was used by Guisado-González et al. [45] using previous waves of the BEEPS data.

For the green innovation, using two direct questions, we verified if the company has introduced new measures to enhance energy efficiency through the first question. Then, using the second question, we examined whether these measures were developed within the company. Hence, we defined a dummy variable that takes "1" if the firm implemented new measures developed by the company itself to enhance the energy efficiency, and "0" otherwise. Green innovation is characterized by higher risk and a longer return on investment period [56]. According to Calogirou et al. [57], SMEs are responsible for almost 64% of the total business ecological impact in the EU. Thus, considering green innovation among the innovation proxies should be of interest since it gives legitimacy.

### 3.2.2. Independent Variable

Informal competition "INF_COM": Based on our data, and following Pérez et al. [7], we estimated the competition from unregistered firms by an ordinal variable that reflects its level. Through answering the question "How Much of An Obstacle: Practices of competitors in informal sector?" a scale from 0 to 4 is presented, with "0" referring to no obstacles are perceived, while "4" refers to a major obstacles.

### 3.2.3. Mediating Variable

Credit constraints "CRED_CONS": we defined credit constraints as a mediating variable to check if it influences the relationship between informal competition and innovation. In prior research, credit access definition is ambiguous [58]. Different approaches to measuring credit access are developed. For instance, credit rationing is the most frequent approach [59]. Nevertheless, according to Farre-Mensa and Ljungqvist [60] credit rationing measures based on accounting data, might not provide accurate information to judge credit access. Credit-constrained firms based on the credit rationing method might comfortably raise debt thanks to their favorable financial leverage and benefit from tax savings.

According to Popov's [30] definition, a firm is considered credit-constrained if it needs a line of credit yet has non, due to financial difficulties. To define this variable, the company was asked whether it has a line of credit without high collateral requirements or not. If the firm has a line of credit then it has credit access. If not, we investigate the reason. If the company has enough internal funds and does not need a loan, then it has credit access. If it needs a line of credit, yet it did not for one due to the high-interest rates, high collateral requirements, complex procedures, the amount or the maturity insufficiency or the fear of being refused, we consider it as credit-constrained. Popov [30] explained that rejection fear is mostly related to an old experience of refusal, which discouraged the firm from asking for credit.

The use of BEEPS data allowed us to distinguish between the credit-tightened companies and the credit-relaxed ones according to the stated reasons. Thus, we defined our mediating variable as a dummy one, which takes "1" if the firm is credit-constrained and "0" if it has credit access.

Financial obstacles "FIN_OBS": For a better grasp of the credit constraints issues for SMEs, we used an ordinal variable based on the question "How Much of an Obstacle: Access to Finance" this variable takes a value between 0 and 4, where 0 means no obstacles are faced, while 4 refers to major obstacles. We tested the effect of informal competition on financial obstacles.

### 3.2.4. Second Stage Moderator

To test our second stage-moderated-mediation, we considered the elaboration of a formalized written business plan as an important variable that moderates the credit constraints innovation link. According to Hlioui and Yousfi [43], the corporate business plan could influence significantly corporate innovation, CSR, and financial resources link. In the same line, Cosci et al. [10] underlined that the bank's evaluation of the companies' business plans while assessing creditworthiness fosters innovation. Hence, we verified whether the SME has established a formalized written business strategy or not through a direct question. Our variable "WRTS" takes 1 if the company answered yes and 0 otherwise.

### 3.2.5. Control Variables

In our study, we controlled for the corporate size, age, and being part of a larger firm. We also control for the different forms of ownership (foreign, state, female, and family). In addition, we included certification-quality possession such as the ISO or MCERTS certification and the CEO characteristics such as gender and experience. Finally, we considered R&D as a variable that influences innovation. Our controls are briefly explained as follows:

Size "Size": we defined the enterprise's size through the natural logarithm of the number of employees. This variable may reflect the level of credit constraints and the ability to face the informal competition since smaller firms have fewer resources and cannot benefit from the economy of scale. Yet, they have better growth potential [61,62].

Age "AGE": we defined age as the natural logarithm of the number of the operating years of the firm [8,62] Age may also reflect the firms' experience and ability to face informal competition [62].

Part of a larger firm "PART_LARG": We checked whether the firm is a part of a larger one based on a direct yes/no question. If the company is part of a larger company, it could have better resources' access, which allows it to have a better competitive position and greater access to financial resources.

Ownership structure: we included different ownership forms following McCann and Bahl [8]. These variables are defined as the percentage of each ownership within the firm. Indeed, the ownership structure affects the corporate behaviors and resource availability. Hence, we considered the foreign ownership "FORG_OWN", state ownership "STAT_OWN", female ownership "FEM_OWN", and family ownership "FAM_OWN".

R&D expenses "R&D": we defined the R&D as a dummy variable that takes "1" if the firm has invested in R&D over the last 3 years and "0" otherwise. This variable is retained by researchers as an innovation input [63,64] that may affect the new product development process.

Top manager characteristics: The CEO experience "CEO_EXP" was measured by the log number of the manager's operating years. When the CEO is well-experienced, he/she has better knowledge and ability to innovate and face illegal rivals. Moreover, the CEO's gender affects his/her behaviors and risk aversion level. According to Boulouta [65], women are more risk-averse. Therefore, they avoid risky investments. Furthermore, they are more socially and environmentally responsible, which might foster green innovation. Hence, we use a dummy variable "FEM_CEO" that takes 1 if the CEO is female.

Internationally recognized quality certification "CERT_QU": According to Chaudhuri et al. [66] possessing a quality certification improves the loan approval likelihood and helps SMEs to have better credit access. Moreover, its effect on innovation capacity has been

widely discussed in prior researche. "CERT_QU" is a dummy variable that takes "1" if the firm has an internationally recognized quality certification and "0" otherwise.

### 3.3. Methodology and Models Specification

The main aim of this research was to explore the conditional indirect effect of credit constraints on informal competition and corporate innovation link while considering the formalized business plan second stage moderating effect. This mediation effect determines the direct and indirect effects of an independent variable on the dependent variable using another mediating variable [67]. The indirect effect explains an independent variable influence on the mediating variable that in its turn affects the dependent variable.

The approach of Baron and Kenny [68] was typically adopted by prior research to test the mediation effect. Yet, Iacobucci et al. [69] demonstrated that using regression separately as suggested by Baron and Kenny could lead to a serious drawback when compared with the structural equation model. Indeed, the separated regression technique produces a larger standard error compared to the structural equation model, which estimates all parameters simultaneously. Zhao et al. [70] supported Iacobucci et al. [69] and considered SME an optimal framework to conduct mediation analysis.

Iacobucci et al. [69] went a step further and suggested the replacement of what was added by Baron and Kenny to the BK approach to Sobel's test with the bootstrap test of the indirect effect. They argue that based on the bootstrap test significance, an indirect effect can be confirmed.

More recently, Hayes [71] introduced the condition of the indirect effect in his book and went beyond the simple mediation modelling to a moderated mediation or mediated moderation. In our study, we investigated a second-stage moderated mediation of the informal competition effect on corporate innovation. Hence, we used a generalized structural equation model with bias-corrected and percentile bootstrap resampling. Our first equation investigated the effect of our independent variable, the informal competition on our mediator the credit constraints:

$$\text{Credit constraints }_{i,j,k} = \alpha + \beta_1 \text{ informal competition }_{i,j,k} + \beta_{i,j} W_{i,j,k} + \varepsilon_{i,j,k} \qquad (1)$$

where i refers to the company, j refers to the country, and k refers to the industry. Credit constraints refer to CRED_CONS and FIN_OBS variables. Informal competition is measured through the INF_COM variable. W includes Size, PART_LARG, FORG_OWN, STAT_OWN, FAM_OWN, AGE, CEO_EXP, FEM_CEO, and CERT_QU. The country and industry fixed effects are controlled. $\varepsilon$ is the equation's error term.

$$\begin{aligned}\text{Innovation}_{i,j,k} = \alpha' + \beta'_1 \text{ informal competition}_{i,j,k} + \beta'_2 \text{ credit constraints}_{i,j,k} + \\ \beta'_3 \text{ Formalized business plan}_{i,j,k} + \beta'_4 \text{ (credit constraints x} \times \text{Formalized} \\ \text{business plan)}_{i,j,k} + \beta'_5 W'_{i,j,k} + \varepsilon'_{i,j,k}\end{aligned} \qquad (2)$$

where Innovation is measured through the 4 innovation proxies defined previously (PROD_INNO; PROC_INNO; RAD_INNO; GREEN_INNO). The formalized business plan is presented by the WRTS variable and W' presents the control variables Size, PART_LARG, FORG_OWN, STAT_OWN, FAM_OWN, AGE, CEO_EXP, FEM_CEO, CERT_QU and R&D. $\varepsilon'$ is the equation's error term.

Through the generalized structural equation model presented in the first and second equations, we define the direct effect of informal competition on innovation through the $\beta'_1$ parameter, while the conditional indirect effect is as following:

$$\text{Conditional indirect effect} = \beta_1 \times (\beta'_2 + \beta'_4 \text{ x} \times \text{Formalized business plan}) \qquad (3)$$

The bootstrap test of the conditional indirect effect focuses on whether the conditional indirect effect is statistically significant based on the bootstrap:

Neither the direct effect nor the bootstrap test of the condition indirect effect is significant: There is no effect non-moderated-mediation.

- The bootstrap test of the conditional indirect effect is not significant. Yet, the direct effect is significant: There is only a direct effect (i.e., no moderated mediation).
- The bootstrap test of the conditional indirect effect is significant, while the direct link is non-significant: there is only a conditional indirect effect (i.e., full moderated mediation).
- Both the bootstrap test of the conditional indirect effect as well as the direct effect are significant:
  - If the coefficients have the same direction: There is complementary moderated mediation.
  - If the coefficients have an opposite direction: There is competitive moderated mediation.

The total effect combine the direct and indirect effect of the informal competition.

## 4. Results

### 4.1. Descriptive Statistics

Figure 1 presents the frequency of our innovation proxies by country. We report that 36% of Eastern European SMEs introduced a new product or significantly enhanced an existing one. This average is higher than the average reported by Ramadani et al. [72] (27.69%) who used the 2013–2014 wave. Thus, an increase in SMEs' product innovation between the two waves is deduced. For process innovation, an average of 22% is reported, while 23.6% of firms underlined that their new products or processes are original and new to the market. We point out that on one hand, Slovenia has the highest average among the Eastern European countries, in terms of product, process, and radical innovation, which is consistent with the office of the European Commission—DG Research and Innovation [73]. In the same line, Crnigoj Marc and Svetin [74] indicated that during 2012–2014 more than 46% of Slovenian enterprises were innovatively active. On the other hand, the Slovak republic registered the lowest averages for product, process, and radical innovation, respectively, with 17%, 7.7%, and 11.3%. Indeed, according to Kohnová [75], SMEs in Slovakia consider knowledge exchange and transfer of knowledge as a threat rather than shared competitiveness, which impedes their innovativeness. Indeed, the Ministry of investments regional development and informatization of the Slovak republic in 2020 underlined that the Slovak government set up an encouraging SMEs' innovation legislation such as the changes in the amendment of the Act 595/2003 on income tax. Although the legislation was intended for SMEs, there is a lack of advantageous incentives, which explains the low level of SMEs' innovation in Slovakia. Besides, according to the European Commission, in 2018, the available SME funding for innovation in the Slovak republic was the lowest among the EU members.

For the green innovation, we registered the lowest frequency among the four innovation types. Less than 20% of the Eastern SMEs declared the energy efficiency measures' development within their companies. Although ecological responsibility is a viral topic in the business field, SMEs experience considerable barriers that prevent tracking green business opportunities [76]. These are related to their resource constraints, experiences, knowledge limitations, and skill deficits. Moreover, SMEs believe that they are partially out of the environmental scope and have a small environmental impact, and thereby they have no issue to consider. Latvia leads the SMEs Green innovation with 38% of companies' developed energy efficiency measures. Indeed, it provides a great ecological collaborative focus to SMEs, for instance, the recent bilateral partnership with Norway for small projects and the DigiBEST project.

Figure 2, as well as Table 2, presents the averages of innovation types by industry. Results illustrate a considerable upper position of the manufacturing sector for all the innovation types. While the textile sector occupies a lesser position in terms of product, process, and radical innovation, it comes in second order in terms of green innovation with a slight setback compared to the manufacturing industry. According to Naudé et al. [77]

since the early stages of the transition period of the Eastern European countries, the overall trend in the manufacturing industry's share in the GDP has grown considerably. The report of the United Nations Geneva [78] constructed a readiness innovation index that includes five building blocks: information communication technology, workforce skills, R&D, industry activity, as well as funding access. According to the index classification, the Eastern European countries have a score under the Western ones, yet higher than the worldwide average. This index spotlights the financial matter that affects considerably SMEs' innovation. In the same line, the European Innovation Scoreboard [79] indicated that in-house innovation by Slovakian SMEs was less than half of the EU average and explained the reason behind this low innovating engagement by the financial gap.

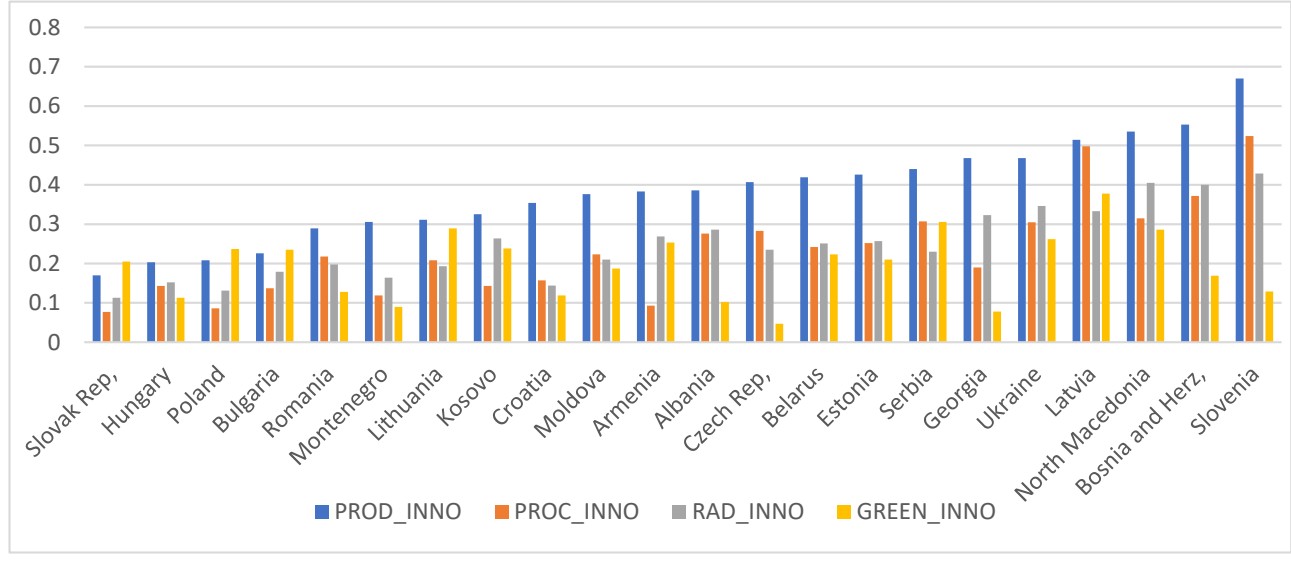

**Figure 1.** The frequency of corporate innovation types by Country.

**Table 2.** The frequency of corporate innovation types by industry.

|  | **PROD_INNO** | **PROC_INNO** | **RAD_INNO** | **GREEN_INNO** |
|---|---|---|---|---|
| Fabricated Metal Products | 0.269 | 0.22 | 0.171 | 0.17 |
| Food & Beverages | 0.341 | 0.207 | 0.247 | 0.206 |
| Garments | 0.382 | 0.188 | 0.288 | 0.25 |
| Hotels | 0.391 | 0.113 | 0.219 | 0.063 |
| Machinery & Equipment | 0.262 | 0.173 | 0.210 | 0.19 |
| Manufacturing | **0.459** | **0.309** | **0.311** | **0.279** |
| Non-Metallic Mineral Products | 0.332 | 0.135 | 0.181 | 0.227 |
| Other Manufacturing | 0.378 | 0.228 | 0.260 | 0.213 |
| Other Services | 0.379 | 0.228 | 0.221 | 0.161 |
| Retail | 0.336 | 0.187 | 0.212 | 0.151 |
| Textiles | **0.134** | **0.059** | **0.142** | 0.277 |

Values presented in bold refers to the maximum and minimum.

Figure 3 presents the credit-constrained firms by country. Our results indicate a difference of 1.5% between the European Union countries and the non-European Union ones. Indeed, EU members face fewer credit constraints. In our sample, Poland has the lowest average (27%). Several agreements can explain the low frequency compared to the other countries. For instance, the Gospodarstwa-Krajowego Bank (BGK) collaboration with the European Investment Fund (EIF) provides additional financial support for SMEs. An agreement was signed in 2020 under the EU SME grantee program COSME.

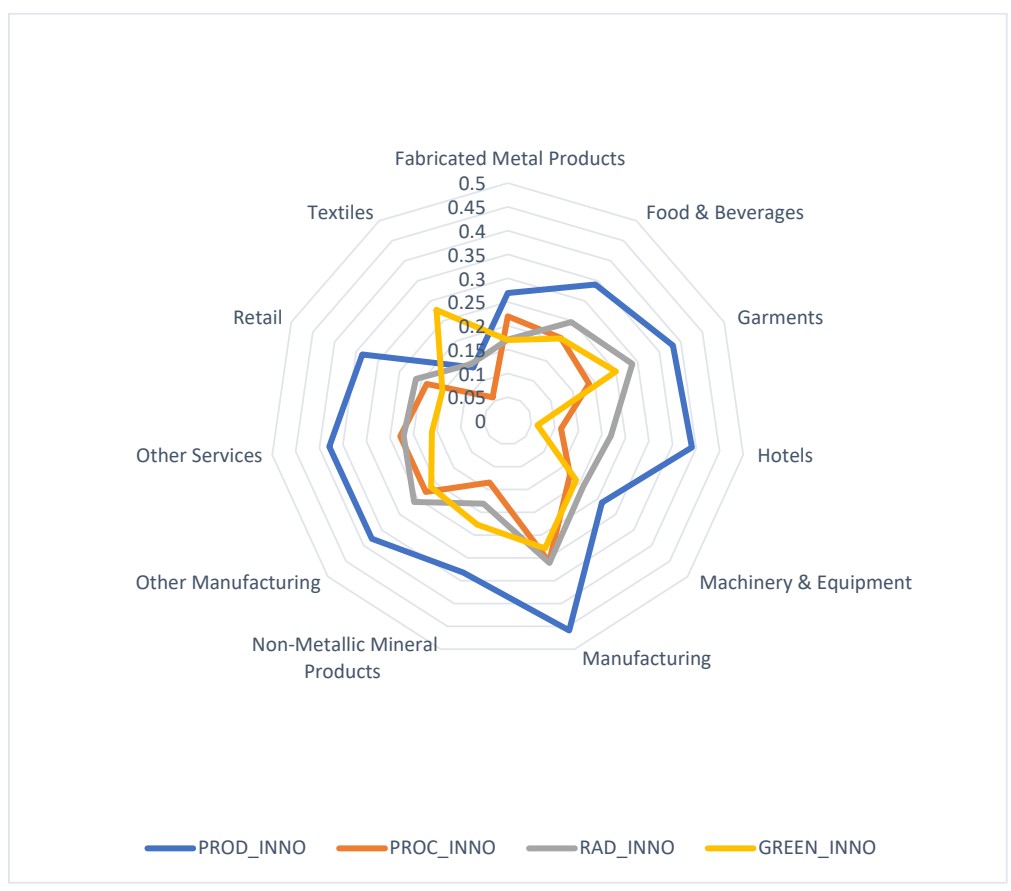

**Figure 2.** Eastern European SMEs innovation by industry.

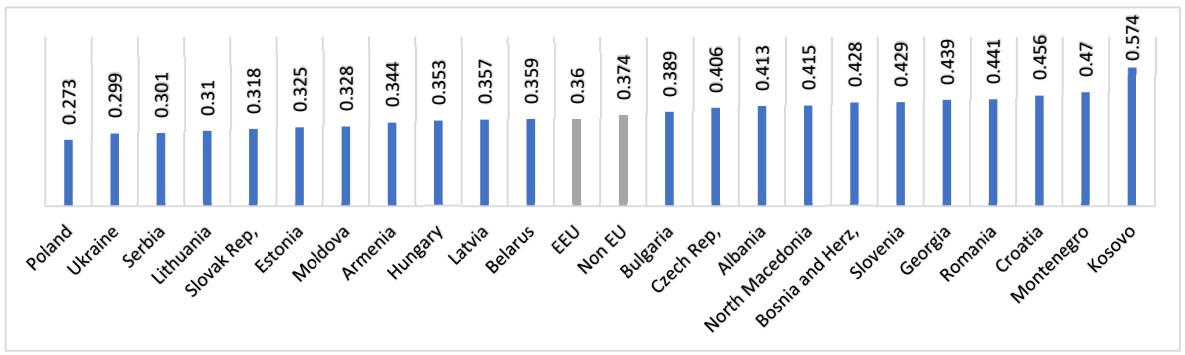

**Figure 3.** Credit constraints frequency by country.

Kosovo has the highest average of constrained SMEs (57%). According to the Assessment of financing needs of SMEs in the Western Balkans countries [80], small and medium loan availability can satisfy the market funding demand. Yet, high collateral requirements are the main obstacle to access to finance and specifically credit availability by SMEs. Despite the fact that banks in Kosovo have more than sufficient liquidity, borrowers need to provide at least guarantee facilities in order to gain access with less conservative collateral conditions.

We note that based on Figure 4 Kosovo has the highest average of informal competition among Eastern European countries. This could be related to the bank collateral requirements, since informal competition threatens the SMEs' yields. The high development of the informal sector in Kosovo is due to the high unemployment rate [80]. In addition, we underscore the considerable difference between the informal competition averages in the EU countries and the non-EU. According to the European Union and OECD (2015), the

formalization strategies to fight against informal economies are based on two ideas, the first is cost-benefit logic, while the second focuses on behavioral change through improving the relationship between governments and entrepreneurs. For instance, the welfare bridges approach used in Croatia, Slovakia, and Hungary, consists of giving a similar equal start-up grant regardless of the received gain.

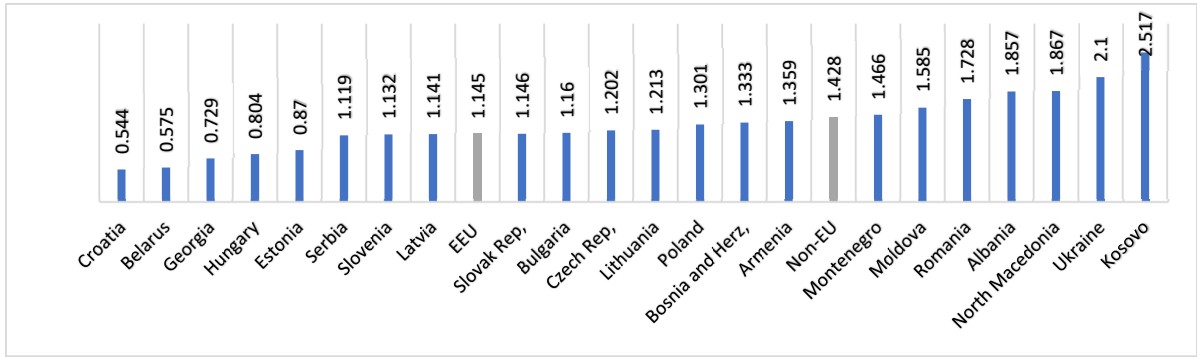

**Figure 4.** Informal competition level by country.

We point out that the average of companies having established business strategies is less the 50%. More than half of Eastern European SMEs operate without a clear business strategy. Figure 5 presents the percentage of companies, which have a formalized written business strategy, by industry. The results confirm that SMEs in the Eastern region are neglecting the elaboration of business plans, which could affect significantly their innovativeness, especially green innovation.

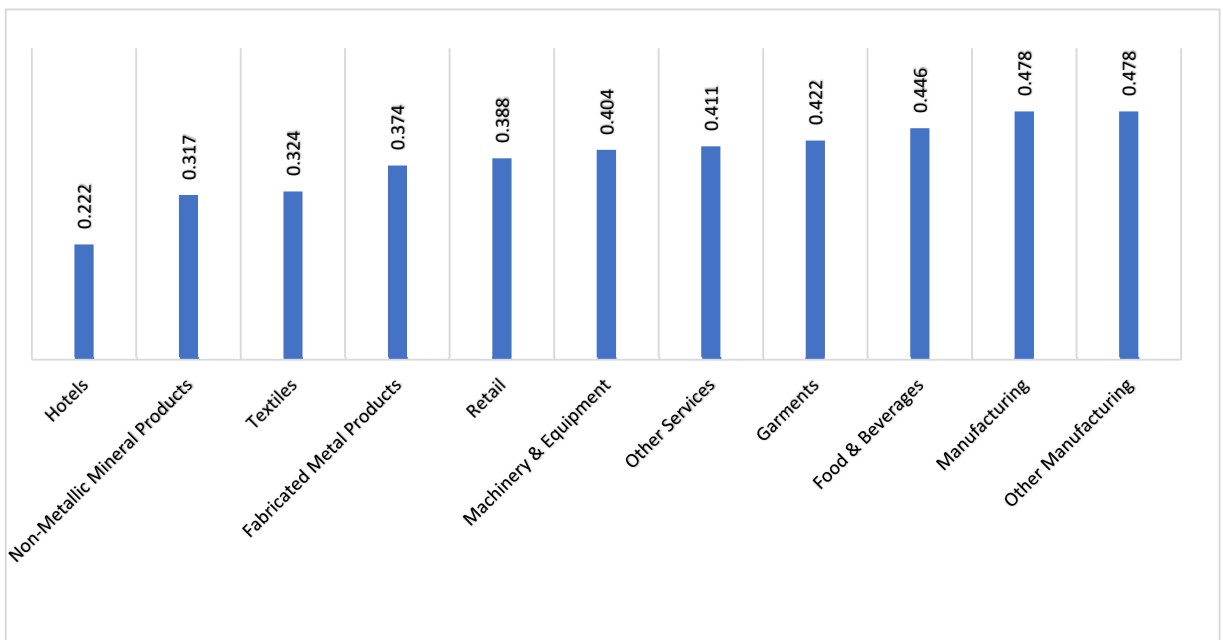

**Figure 5.** Frequency of companies with formalized business strategies by industry.

Figure 6 illustrates the percentage of SMEs that have introduced PROD_INNO, PROC_INNO, RAD_INNO and GREEN_INNO. We notice an increased frequency of SMEs' innovation based on the informal competition level, except for green innovation when the informal competition leads to very severe obstacles. Hence, we assume that those firms are using innovation to differentiate from their informal rivals and adapt to their framework. Table 3 presents the pairwise correlation matrix. It confirms the absence of high correlation between the independent variables.

**Table 3.** Correlation Matrix.

| Variables | (1) | (2) | (3) | (4) | (5) | (6) | (7) | (8) | (9) | (10) | (11) | (12) | (13) | (14) | (15) | (16) | (17) | (18) |
|---|---|---|---|---|---|---|---|---|---|---|---|---|---|---|---|---|---|---|
| (1) PROD_INNO | 1.000 | | | | | | | | | | | | | | | | | |
| (2) PROC_INNO | 0.315 *** | 1.000 | | | | | | | | | | | | | | | | |
| (3) RAD_INNO | 0.446 *** | 0.305 *** | 1.000 | | | | | | | | | | | | | | | |
| (4) GREEN_INNO | 0.129 *** | 0.147 *** | 0.132 *** | 1.000 | | | | | | | | | | | | | | |
| (5) INF_COM | 0.050 *** | 0.076 *** | 0.056 *** | 0.048 *** | 1.000 | | | | | | | | | | | | | |
| (6) Credit_Const | −0.107 *** | −0.105 *** | −0.097 *** | −0.079 *** | 0.054 *** | 1.000 | | | | | | | | | | | | |
| (7) Size | 0.121 *** | 0.174 *** | 0.108 *** | 0.154 *** | −0.053 *** | 0.194 *** | 1.000 | | | | | | | | | | | |
| (8) PART_LARG | 0.023 ** | 0.060 *** | 0.014 | 0.025 ** | −0.053 *** | 0.026 ** | 0.116 *** | 1.000 | | | | | | | | | | |
| (9) AGE | 0.014 | 0.042 *** | 0.013 | 0.071 *** | 0.001 | 0.050 *** | 0.175 *** | 0.032 *** | 1.000 | | | | | | | | | |
| (10) CEO_EXP | 0.011 | 0.025 ** | 0.018 * | 0.026 ** | −0.014 | −0.001 | 0.003 | −0.009 | 0.163 *** | 1.000 | | | | | | | | |
| (11) FEM_CEO | −0.002 | −0.035 *** | −0.005 | −0.039 *** | 0.007 | −0.059 *** | −0.094 *** | 0.027 ** | −0.023 ** | −0.012 | 1.000 | | | | | | | |
| (12) CERT_QU | 0.026 ** | 0.040 *** | 0.020 * | 0.041 *** | −0.025 ** | 0.045 *** | 0.078 *** | 0.006 | 0.061 *** | −0.042 *** | −0.042 *** | 1.000 | | | | | | |
| (13) FORG_OWN | 0.057 *** | 0.077 *** | 0.062 *** | 0.037 *** | −0.074 *** | −0.019 * | 0.217 *** | 0.093 *** | −0.042 *** | −0.042 *** | 0.027 ** | 0.027 ** | 1.000 | | | | | |
| (14) STAT_OWN | −0.015 | −0.022 ** | −0.024 ** | 0.013 | −0.060 *** | 0.008 | 0.105 *** | 0.034 *** | 0.084 *** | −0.016 | 0.027 ** | 0.004 | −0.079 *** | 1.000 | | | | |
| (15) FAM_OWN | 0.052 *** | 0.038 *** | 0.041 *** | 0.023 ** | −0.006 | 0.017 | −0.146 *** | −0.001 | 0.069 *** | 0.041 *** | 0.059 *** | 0.031 ** | −0.028 ** | −0.107 *** | 1.000 | | | |
| (16) R&D | 0.339 *** | 0.299 *** | 0.326 *** | 0.188 *** | 0.073 *** | 0.108 *** | 0.208 *** | 0.023 ** | 0.015 | 0.017 | −0.058 *** | 0.051 *** | 0.075 *** | −0.032 *** | 0.011 | 1.000 | | |
| (17) EU | 0.073 *** | 0.017 | 0.080 *** | 0.006 | −0.046 *** | −0.009 | −0.026 ** | −0.029 *** | 0.098 *** | 0.012 | 0.057 *** | 0.031 *** | 0.010 | −0.090 *** | 0.169 *** | −0.085 *** | 1.000 | |
| (18) WRTS | 0.156 *** | 0.184 *** | 0.167 *** | 0.154 *** | −0.017 | −0.086 *** | −0.307 *** | −0.111 *** | −0.067 *** | 0.010 | 0.005 | −0.052 *** | −0.142 *** | −0.096 *** | 0.045 *** | −0.239 *** | 0.074 *** | 1.000 |

*** $p < 0.01$, ** $p < 0.05$, * $p < 0.1$.

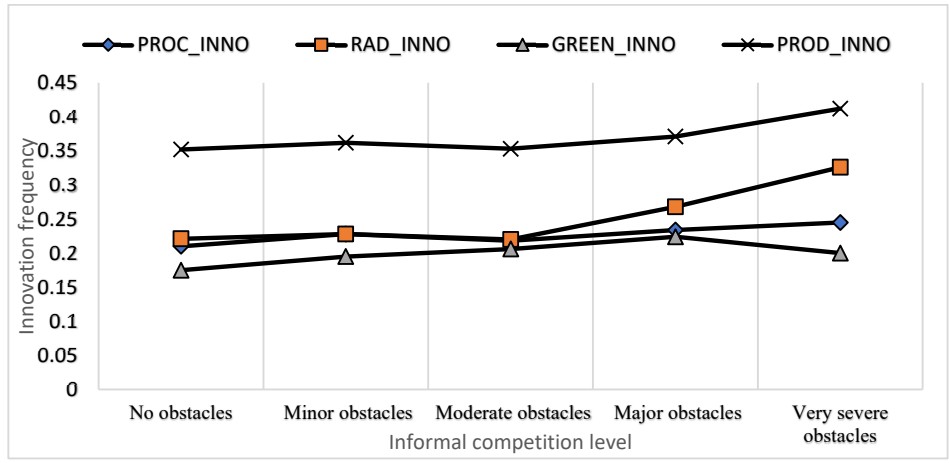

**Figure 6.** The innovation frequency by informal competition level.

*4.2. Results*

To test the conditional indirect effect of the informal completion–innovation link mediated by the credit constraints and the second stage moderated by the business strategy elaboration, we applied a generalized structural equation model with a bias-corrected and percentile bootstrap. Table 4 presents the effect of informal competition on credit constraints. Our results reveal a positive effect of informal competition on credit constraints and financial obstacles' likelihood. Hence, our second assumption is accepted.

The same table shows a negative effect of foreign ownership and family ownership on both credit constraints and financial obstacles. Indeed, in line with Dalgıç and Fazlıoğlu [55], the presence of foreign owners provides other financial alternatives that might relax the financial barriers. Moreover, the female CEO's presence has a negative effect on the credit constraints, while it does not affect significantly the financial obstacles. This could be explained by the fact that lenders tend to require stricter conditions when it comes to female managers. Yet, to overcome the credit constraint, women can benefit from other financial support programs. Hence, they do not face generally though financial obstacles. Furthermore, the CEO experience helps to lessen financial obstacles. Well-experienced CEOs have better knowledge and a generally larger business network. They know how to deal with their surroundings.

**Table 4.** Informal competition effect on credit constraints and financial obstacles.

|  | FIN_OBS | Credit_Const |
| --- | --- | --- |
| INF_COM | 0.468 *** | 0.126 *** |
|  | (0.019) | (0.020) |
| EU | 0.868 *** | 0.363 ** |
|  | (0.171) | (0.184) |
| PART_LARG | −0.031 | 0.073 |
|  | (0.083) | (0.088) |
| FORG_OWN | −0.004 *** | −0.006 *** |
|  | (0.001) | (0.001) |
| STAT_OWN | −0.002 | −0.002 |
|  | (0.003) | (0.003) |
| FAM_OWN | −0.001 ** | −0.002 ** |
|  | (0.001) | (0.001) |
| AGE | −0.094 *** | 0.058 |
|  | (0.036) | (0.042) |
| CEO_EXP | −0.089 *** | 0.014 |
|  | (0.033) | (0.039) |
| FEM_CEO | −0.052 | −0.160 ** |
|  | (0.057) | (0.065) |

**Table 4.** *Cont.*

|  | **FIN_OBS** | **Credit_Const** |
|---|---|---|
| Size | 0.018 (0.022) | 0.420 *** (0.025) |
| CERT_QU | 0.017 (0.018) | 0.047 ** (0.021) |
| Industry | Yes | Yes |
| Country | Yse | Yse |
| Number of obs | 7599 | 7622 |
| Family | ordinal | Bernoulli |
| Link | logit | logit |

The table presents the effect of informal completion on the log odds ratio of being credit-constrained using a generalized structural equation model GSEM logit estimation. It shows the informal completion effect on the financial obstacles level through a generalized structural equation model ordered logit estimation. The rest of the GSEM is presented in Table 5. Values between the parentheses refer to the standard errors. The country and industry fixed effects are controlled. The stars indicate the significance level with *** $p < 0.01$, ** $p < 0.05$.

**Table 5.** Conditional indirect effect of informal competition on innovation through credit constraints mediation and formalized business strategy's second-stage moderation.

|  | **PROD_INNO** | **PROC_INNO** | **RAD_INNO** | **GREEN_INNO** |
|---|---|---|---|---|
| CRED_CONS | −0.260 *** (0.036) | −0.244 *** (0.046) | −0.286 *** (0.042) | −0.352 *** (0.056) |
| WRTS X CRED_CONS | −0.037 ** (0.014) | −0.045 ** (0.018) | −0.040 ** (0.016) | 0.301 (0.213) |
| WRTS | 0.438 *** (0.058) | 0.561 *** (0.068) | 0.609 *** (0.065) | 0.532 *** (0.089) |
| INF_COM | 0.082 *** (0.021) | 0.161 *** (0.025) | 0.074 *** (0.024) | 0.086 *** (0.026) |
| EU | 1.015 *** (0.193) | 1.035 *** (0.204) | 0.306 (0.197) | 0.049 (0.310) |
| PART_LARG | 0.047 (0.091) | 0.198 ** (0.092) | −0.089 (0.103) | 0.184 * (0.111) |
| FORG_OWN | 0.001 (0.001) | 0.001 (0.001) | 0.003 ** (0.001) | 0.003 ** (0.001) |
| STAT_OWN | −0.010 *** (0.003) | −0.014 *** (0.005) | −0.013 *** (0.004) | −0.001 (0.004) |
| FAM_OWN | 0.005 *** (0.001) | 0.004 *** (0.001) | 0.004 *** (0.001) | 0.004 *** (0.001) |
| CEO_EXP | 0.038 (0.040) | 0.089 * (0.050) | 0.052 (0.046) | 0.168 ** (0.052) |
| FEM_CEO | 0.050 (0.066) | −0.192 ** (0.082) | 0.050 (0.074) | −0.263 ** (0.086) |
| R&D | 1.561 *** (0.053) | 1.490 *** (0.057) | 1.587 *** (0.056) | 0.820 *** (0.065) |
| CERT_QU | 0.0428 ** (0.018) | 0.092 *** (0.026) | 0.039 * (0.021) | 0.058 ** (0.027) |
| AGE | −0.048 (0.043) | −0.095 * (0.051) | −0.003 (0.048) | 0.076 (0.054) |
| Size | 0.143 *** (0.027) | 0.291 **** (0.032) | 0.104 *** (0.030) | 0.245 *** (0.034) |
| Industry | Yes | Yes | Yes | Yes |
| Country | Yes | Yes | Yes | Yes |
| Number of obs | 7464 | 7437 | 7439 | 7330 |

Using a structural equation model we estimated the informal competition direct and indirect effect on innovation. Values between the parentheses refer to the standard errors. The country and industry fixed effects are controlled. The stars indicate the significance level with *** $p < 0.01$, ** $p < 0.05$, * $p < 0.1$.

Table 5 presents the conditional indirect effect of informal competition on the innovation likelihood through credit constraints' mediation and business strategy's second-stage moderation. Results show a negative association between credit constraints and all the innovation types. Hence, our third assumption is accepted. Innovative small businesses need external financial support to innovate. Moreover, the credit constraints' effect is more pronounced on the green innovation likelihood. This innovation type is riskier and requires a long payback period, thereby lenders are stricter, which might impede SMEs' environmental innovation.

Regarding the direct effect of informal competition on corporate innovation, we register a positive significant coefficient for all the innovation types. SMEs are trying to innovate to provide inimitable products and processes, which helps to overcome informal competitors. Our first assumption is accepted. This positive effect is in line with Pérez et al. [7] and Miocevic et al. [28]. We note that informal competition fosters process innovation the most among the innovation types. Since implementing new processes depends on specific conditions and frameworks, it cannot be easily mitigated. Therefore, process innovation could present an interesting tool for diversification. For the formalized business strategy establishment, our results highlight a positive effect on all the corporate innovation categories. Moreover, a significant negative joint effect of the credit constraint and formalized business strategy development is registered for the product, process and radical innovation.

Other than the credit constraints and informal competition, we find a positive significant effect of the R&D expenses as well as the international certification quality on all the innovation types. We also note that the CEO experience enhances green innovation since it provides better knowledge, while the European Union membership has a positive significant effect registered on product and process innovation. Besides, larger companies benefit from the economy of scale, which fosters innovation capacity.

Table 6 presents the bootstrap results for the conditional indirect effect. According to our results, the direct effect of informal competition fosters all innovation proxies. Yet, since informal competition increases the credit constraints effect, which in its turn reduces innovation, a negative indirect effect is registered. This indirect effect is moderated by the business plan establishment. Indeed, the credit constraints' negative effect on innovation is lessened by the formalized business strategy development. Yet, for green innovation, the second-stage moderating effect is non-significant. Our bootstrap results confirm the significant conditional indirect effect for all the innovation categories. Hence, the fourth assumption is accepted. There is a competitive moderated mediation.

**Table 6.** Bootstrap results.

| _bs | Coef. | [95%Conf.Interval] | | |
|---|---|---|---|---|
| | | **Number of Obs = 7795** | | |
| | | **Replications = 1000** | | |
| PROD_INNO | −0.034 *** (0.008) | −0.050 | 0.018 | NB |
| | | −0.051 | 0.019 | P |
| | | −0.052 | 0.019 | BC |
| PROC_INNO | −0.033 *** (0.009) | −0.050 | 0.016 | NB |
| | | −0.053 | 0.017 | P |
| | | −0.053 | 0.018 | BC |
| RAD_INNO | −0.034 *** (0.009) | −0.052 | 0.017 | NB |
| | | −0.053 | 0.018 | P |
| | | −0.055 | 0.021 | BC |
| GREEN_INNO | −0.028 *** (0.008) | −0.043 | −0.012 | NB |
| | | −0.044 | −0.013 | P |
| | | −0.46 | −0.015 | BC |

(NB) normal-based, (P) percentile confidence interval, (BC) bias-corrected confidence interval. Values between the parentheses refer to the standard errors. The stars indicate the significance level with *** $p < 0.01$.

## 5. Discussion

Eastern Europe is a region with impressive growth potential. It has a high business attractiveness and provides great opportunities in technological sectors as fruit of the recently established strategies for innovation improvement as well as the region's features. The educated workforces, cost competitiveness, as well as access to Western Europe are some advantages that foster innovation, especially in the industry 4.0 ecosystem. Since 2004, the admittance of several Eastern European economies to the European Union has increased the delocalization pace in the Western members. Industries started moving from the old member states to the new ones [81], which boost corporate innovativeness in the region. Yet, despite the opportunities, the establishment of a new economic plan that fosters innovation in sluggish context is still challenging. In our study, we focused on some important factors that affect SMEs' innovation, which are credit access and informal competition.

On the basis of our findings, we concluded that, in Eastern Europe, the presence of a significant portion of the informal sector in the economy reduces the formal SMEs' market share and threatens business. Consequently, their repayment capacity shrinks. Companies are more likely to be credit-constrained as the informal economy intensifies. Given that credit constraints are also classified among the main obstacles faced by SMEs in the region, controlling the informal competition could indirectly reduce and lessen these constraints. Moreover, the informal sector includes generally unemployed and low-skilled workers that are less likely to invent, which makes innovation a great source of comparative advantage creation. These results support Lenox and Toffel's [26] statements that companies, specifically SMEs in our study, are subjected to path dependencies that define their competitive advantage.

We also underlined that formalized business strategies enhance all innovation types and decrease the credit constraints' negative effect. The process of preparing a clear formalized business plan helps the company to analyze its surrounding, the possible opportunities and threats, as well as its strengths and weakness. In addition, it could improve understanding of the stakeholders' needs and expectations, which mitigates the innovation risk and grants lenders' trust. Furthermore, we confirm that green innovation is the most influenced by credit constraints. The European Commission [76] has indicated that until recently, the focus of policy-makers and practitioners to foster pro-environmental innovations has been on large firms. Consequently, there was not enough attention when developing strategies to fund SMEs' green innovation. Recently the support mechanisms 2021–2027 introduced within the eco-innovation action plan of the European Union took better consideration of SMEs' participation. The SMEs' financial hurdles are not only due to external factors. Indeed, according to the Flash Eurobarometer Survey, the lack of financial knowledge and the consideration of environmental engagement as an extra activity excluded from the core of the business strategy development are among the main barriers that hinder innovation. Considering Machová et al. [24] study, in which they highlighted that environmental protection is not the most important factor and just a secondary option for Eastern European consumers, SMEs' repayment ability will not improve due to green innovative products. In other words, credit-constrained SMEs have neither sufficient financial resources nor lenders' support to invest in innovative projects and they lack the necessary financial knowledge to seize funding opportunities. This explanation is in line with the significant effect of the CEO's experience only on green innovation. Consistent with the findings of other articles (e.g., [26]) in this special issue and its overall theme, ecologically innovative SMEs face more frequent and intense knowledge barriers than technologically (product, process) innovative SMEs, due to the higher complexity of the knowledge demanded to ecologically innovate.

## 6. Conclusions

Our research aimed to highlight the influence of informal competition on SMEs' innovation in the Eastern European region and whether the credit constraints can mediate this relation while considering a second-stage moderation of business strategies' development.

To reach our aim, we used four innovation proxies, which are product innovation, process innovation, radical innovation, and green innovation. Using the sixth wave of the BEEPS survey as the main data source. Our results indicate that informal competition has a positive direct effect on innovation proxies. Innovation is a tool to stand out in presence of informal competitors. Moreover, the informal sector hinders SMEs' credit access likelihood. In turn, being credit-constrained leads to less corporate innovation. This negative indirect effect is moderated by formalized business strategy development. A partial conditional mediation effect of credit constraints was confirmed for all the innovation types. Innovation requires resource availability, skill, and knowledge. Therefore, elaborating formalized business strategy helps to satisfy the requirements and lessen the credit constraints.

The informal sector reduces the formal companies' market share, which affects profitability, repayment abilities, and lending activity. Yet, it has a positive effect on SMEs' innovation. Hence, regulators should control the informal activity while encouraging formal companies to innovate due to other motivations. The main reason for informal sector development is the unemployment rate and the presence of an inadequately educated workforce that finds the informal economy as an alternative. Hence, programs and opportunities for the unemployed to receive adequate training consistent with the markets' needs and which develop the workforce creativity should limit the informal competition while maintaining innovativeness. In addition, given the moderating effect of elaborating business strategies and the remarkable low average of SMEs that prepare formalized business strategies, it would be better for those firms to follow business financial literacy programs. Our study presents some limits. The binary consideration of the innovation measures and the credit constraints proxy cannot reflect the different levels and importance of the SMEs' innovative efforts. Therefore, future studies could examine whether the conditional indirect link of the informal competition affects innovation intensity.

**Author Contributions:** We declare that all the authors contributed to this research. The original draft was provided by Z.H., the empirical corrections were applied by M.G. and the theoretical improvements were done by A.O. All authors have read and agreed to the published version of the manuscript.

**Funding:** This research received no external funding.

**Institutional Review Board Statement:** Not applicable.

**Informed Consent Statement:** Not applicable.

**Data Availability Statement:** Data are available online in the BEEPS site: https://www.beeps-ebrd.com/data/ (accessed on 9 October 2022).

**Conflicts of Interest:** The authors declare no conflict of interest.

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
