# Peer review of "Informal Competition Effect on SMEs’ Innovation: Do Credit Constraints Matter? Evidence from Eastern European Countries"

_sustainability, doi:10.3390/su142113874_

Round 1

Reviewer 1 Report

Dear authors,
Thank you very much for sending your paper to the journal. The topic is quite interesting, however, the whole paper still is immature to get acceptance. Please kindly incorporate all the following comments in the next version and resubmit as soon as possible:
1-What is the research gap? In other words, why did you select the current study and what is the motivation of the study?
2-The theoretical does not well support the research hypotheses so you need to improve this paper as much as possible.
3-The main issue of the current paper is the research method.
3-1-Why did you select the period from 2018 to 2020? Any specific reasons?
3-2-Why and how did you select 9146 firms?
3-3- could not find your mentioned databases on the web, so please give the direct link to the database in the footnote.
3-4-It is strongly recommended to conduct additional analyses of the study
4-Please add the discussion section to the paper.
5- It is also strongly recommended to conduct proofreading on the paper.

Author Response

Dear Reviewer,

We would like to thank you for your valuable efforts, and for the precious time, you spent on our work. We are grateful for your considerable remarks that aim at improving our manuscript.

In response to your insightful comments, we reported several modifications in order to make the equations as clearer as possible.

In this reply letter, we provide you point-by-point responses.

Please note that paragraphs written in italic are extracted from the paper.

The modifications are the result of all the reviewers' and editor's comments.

Best regards,

Comments:   

Reviewer 1 :

Thank you very much for sending your paper to the journal. The topic is quite interesting, however, the whole paper still is immature to get acceptance. Please kindly incorporate all the following comments in the next version and resubmit as soon as possible:
1-What is the research gap? In other words, why did you select the current study and what is the motivation of the study?

After correcting our paper, we have underlined the research gap and the added value of the paper: for instance, we added:  “To our knowledge, there is no previous study that has examined this conditional indirect link in the eastern European context”  

2-The theoretical does not well support the research hypotheses so you need to improve this paper as much as possible.

Given your recommendations as well as the editor's suggestions, we have improved the literature review. we defined the innovation proxies in this section (as recommended by the 3rd reviewer) while providing a more adequate theoretical background ( based on the editor's suggested papers). 

3-The main issue of the current paper is the research method.

We have modified the research method: A modification of the studied mediation effect to a conditional indirect effect (moderated mediation ) as suggested by the 3rd reviewer ( using business plan moderating effect) 

3-1-Why did you select the period from 2018 to 2020? Any specific reasons?

Thank you for your comment. We have justified the reason behind this period 

“In this paper, we examine the effect of informal competition on innovation from 2018 to 2020 (just before the pandemic outbreak). Indeed, in 2017 some central and eastern European leaders gathered in Warsaw, and underlined the importance of a replacement for the outdated growth model and asked how to establish the new model [1]. Hence, we select the post-gathering period in our study to help answer an urgent need in this context.”    
3-2-Why and how did you select firms?

Please check the modified sample and data section, where we have presented details on the firms’ selection

3-3- could not find your mentioned databases on the web, so please give the direct link to the database in the footnote.

Thank for the remark, Done
3-4-It is strongly recommended to conduct additional analyses of the study

We have changed the used approach and conducted a moderated mediation study using the GSEM with bootstrap resampling test rather than the Baron and Kenny approach to have more accurate results.

4-Please add the discussion section to the paper.

Done, we distinguish between the results and discussion section in the new version. 

Reviewer 2 Report

I enjoyed reading your paper. Here are some comments that may help you improve your paper:

1. Hypotheses: you break down innovation into product, process, radical, and green innovations. This is good, since you can present clearer theory. However, it is also challenging, since you have to justify the arguments. You may improve your arguments for H1, H1a-H2a. You may consider using innovation as the concept, and use types of innovations as robust test. This will simplify your argument. 

2. Mediation argument is relatively weak. You can strengthen that part. 

3. Control variables: It is important to justify your control variables, since control variables may affect your empirical findings (Li, 2021).

4. Table: please align numbers and texts for Tables 3. Please double check formats and numbers in each table to make sure they are correct. 

5. Mediation: for mediation (partial or full mediation), it is important to test percentage of direct effects is mediated by the mediator (Blount & Li, 2021). 

Overall, I hope you will find my comments useful. Good luck. 

Blount, I., & Li, M. (2021). How Buyers' Attitudes Toward Supplier Diversity Affect Their Expenditures with Ethnic Minority Businesses. Journal of Supply Chain Management. 

Li, M. (2021). Uses and abuses of statistical control variables: Ruling out or creating alternative explanations? Journal of Business Research, 126, 472-488.  

Author Response

Dear reviewer,

We would like to thank you for your valuable efforts, and for the precious time you spent on our work. We are grateful for your considerable remarks that aim at improving our manuscript.

In response of your insightful comments, we reported several modifications in order to make the equations as clearer as possible.

In this reply letter, we provide you point-by-point responses.

Please note that paragraphs written in italic are extracted from the paper.

Best regards,

comment:

We note that the studied link is little modified as requested by one of the reviewers  

. Hypotheses: you break down innovation into product, process, radical, and green innovations. This is good, since you can present clearer theory. However, it is also challenging, since you have to justify the arguments. You may improve your arguments for H1, H1a-H2a. You may consider using innovation as the concept, and use types of innovations as robust test. This will simplify your argument. 

I would like to thank you for your recommendation. Combining this comment with the third reviewer's comment, we modified the studied link. We use a conditional indirect effect with bootstrap resampling to check for the robustness of our study.

2. Mediation argument is relatively weak. You can strengthen that part. 

we have modified the mediation section. we hope the new version satisfies the requirements

3. Control variables: It is important to justify your control variables, since control variables may affect your empirical findings (Li, 2021).

thank you for your recommendation. We provide a brief justification for the control variables while conserving moderated length of the paper.

4. Table: please align numbers and texts for Tables 3. Please double check formats and numbers in each table to make sure they are correct. 

We have represented the 3rd table while considering the newly added variables.

5. Mediation: for mediation (partial or full mediation), it is important to test percentage of direct effects is mediated by the mediator (Blount & Li, 2021).

we use the bootstrap test for conditional mediation to test the significance of the conditional indirect effect.

Reviewer 3 Report

The paper investigates an interesting issue and is well presented. However, I would suggest to the authors give a brief of some other countries' experiences. Please see below some suggested recently published papers:

Dalgıç, B. & Fazlıoğlu, B. (2021). Innovation and firm growth: Turkish manufacturing and services SMEs. Eurasian Business Review, 11(3), 395–419.

Molodchik, M., Jardon, C. & Yachmeneva, E. (2021). Multilevel analysis of knowledge sources for product innovation in Russian SMEs. Eurasian Business Review, 11(2), 247–266.

Also, I would suggest to the authors compare their results with the results of the previous studies.

Moreover, the motivation and contribution of the paper should be written clearly.

Author Response

Dear reviewer,

We would like to thank you for your valuable efforts, and for the precious time you spent on our work. We are grateful for your considerable remarks that aim at improving our manuscript.

In response of your insightful comments, we reported several modifications in order to make the equations as clearer as possible.

In this reply letter, we provide you point-by-point responses.

Please note that paragraphs written in italic are extracted from the paper.

Best regards,

The paper investigates an interesting issue and is well presented. However, I would suggest to the authors give a brief of some other countries' experiences. Please see below some suggested recently published papers:

Dalgıç, B. & Fazlıoğlu, B. (2021). Innovation and firm growth: Turkish manufacturing and services SMEs. Eurasian Business Review, 11(3), 395–419.

Molodchik, M., Jardon, C. & Yachmeneva, E. (2021). Multilevel analysis of knowledge sources for product innovation in Russian SMEs. Eurasian Business Review, 11(2), 247–266.

Thank you for the suggested paper. We considered these studies to strengthen our theoretical background. 

Also, I would suggest to the authors compare their results with the results of the previous studies.

Thank you for your recommendation after the modification we have compared the study with some converging investigation.

Moreover, the motivation and contribution of the paper should be written clearly.

We tried in the new version of this paper to highlight the research gap and the contribution of our study. 

Reviewer 4 Report

Thanks for allowing me to review this research. This study tends to duplicate previous studies of how informal competition works on innovations but switches to the regional level with the context of Eastern European countries.

Introduction:

My biggest theoretical concern is related to the research gaps the authors didn’t identify. The paper tends to integrate three concepts, financial constraints, informal competition, and innovation, but without an overall theoretical framework. A lack of arguments of “why this study focuses on the topic” is missing. In my opinion, just switching the context and using the data from east Europe can be a convincing research gap. Why did the study decide to create the mediation relationship? Did previous studies didn’t investigate already? What’s new about this study? 

Literature review:

The authors didn’t offer a clear understanding of the literature review on any of the three concepts. Simply listing some studies from each concept, instead of showing the connections and rationales of the connections, would just miss the purpose of the literature review. I am highly concerned about why the authors skipped the entire literature review and jumped into the hypothesis development directly.

In addition, the study didn’t provide the definitions of the four innovations until the section on measurements. It’s so hard to follow the hypothesis development since these statements could easily be used for any of these hypotheses if changed the names of the innovations are.  

Furthermore, the study didn’t cite appropriately all over the paper. Many key arguments have no reference, making not only the literature review unclear but also the introduction vague.

Materials and Methods:

The study didn’t explain well why this sample and data were used rather than other data. What are the advantages of the sample especially for this study?

Using dummy as the DVs hurts the study a lot. Is that possible that innovation refers to both product and process innovation? The dummy creates more confusion for radical and green innovation. The two concepts can be overlapped in many cases.

Also, the approach to medication effect by Baron and Kenny is outdated. Please consider the full mediation model using a conditional indirect effect analysis with bias-corrected bootstrap resampling by Hayes (2017)’s PROCESS procedure.

It’s disappointed that the study ignores the high correlations in table 2 without any explanations. It created a huge issue of multicollinearity, which makes the following analyses meaningless. Moreover, the correlation table didn’t show the correlations among the four DVs, which I am highly concerned about. Based on the descriptions of the first three DVs, I couldn’t find a clear difference. 

Conclusions:

It appears that the conclusion just summarized the results and didn’t contain any specific convinced contributions.

Author Response

Dear reviewer,

We would like to thank you for your valuable efforts, and for the precious time you spent on our work. We are grateful for your considerable remarks that aim at improving our manuscript.

In response of your insightful comments, we reported several modifications in order to make the equations as clearer as possible.

In this reply letter, we provide you point-by-point responses.

Please note that paragraphs written in italic are extracted from the paper.

Best regards,

Comments:   

            Thanks for allowing me to review this research. This study tends to duplicate previous studies of how informal competition works on innovations but switches to the regional level with the context of Eastern European countries.

Introduction:

My biggest theoretical concern is related to the research gaps the authors didn’t identify. The paper tends to integrate three concepts, financial constraints, informal competition, and innovation, but without an overall theoretical framework. A lack of arguments of “why this study focuses on the topic” is missing. In my opinion, just switching the context and using the data from east Europe can be a convincing research gap. Why did the study decide to create the mediation relationship? Did previous studies didn’t investigate already? What’s new about this study? 

We have presented several modifications based on the presented questions. For instance:

“To our knowledge, there is no previous study that has examined this conditional indirect link in the eastern European context. While studies about the separated links are well developed in the literature there was no previous attempt to investigate the conditional indirect link.   “

Literature review:

The authors didn’t offer a clear understanding of the literature review on any of the three concepts. Simply listing some studies from each concept, instead of showing the connections and rationales of the connections, would just miss the purpose of the literature review. I am highly concerned about why the authors skipped the entire literature review and jumped into the hypothesis development directly. In addition, the study didn’t provide the definitions of the four innovations until the section on measurements. It’s so hard to follow the hypothesis development since these statements could easily be used for any of these hypotheses if changed the names of the innovations are.  

we have rewritten the literature review section while considering your comment. please check the new version of the paper attached the letter.

Furthermore, the study didn’t cite appropriately all over the paper. Many key arguments have no reference, making not only the literature review unclear but also the introduction vague.

We modified the literature review section based on your recommendation as well as the other reviews recommendation. As recommended, We present a brief definition, in the literature review section

Materials and Methods:

The study didn’t explain well why this sample and data were used rather than other data. What are the advantages of the sample especially for this study?

This survey provides financial, environmental, investment and general data for SMEs. Since generally it is hard to find published financial and/or extra financial statements for SMEs survey data can provide information through direct questions. Therefore, the truthfulness of the  Perception of The Questions Regarding Opinions is required the BEEPS data provides a distinction between truthful answers and doubtful ones.  

We modified the sample and data section to highlight the advantages of our sample.

Using dummy as the DVs hurts the study a lot. Is that possible that innovation refers to both product and process innovation? The dummy creates more confusion for radical and green innovation. The two concepts can be overlapped in many cases.

Thank you for your recommendation. We modified the innovation variables description to make it clearer.

…, we consider product innovation if there is a new product development and process innovation if the company indicated that it has developed a new process. We define these two categories based on two direct questions, in which the firm answers whether:

 If the firm has introduced a new product or significantly enhanced an existing one over the last three fiscal years, then it has product innovation. For the second question, has the company introduced a new process or significantly enhanced an existing one, then it has process innovation.

Hence, we introduce two dummy variables. Each variable takes “1” if the answer to its related question is yes and “0” otherwise …”

Indeed, to measure innovation there are two main approaches. The first is through innovation input (R&D). While the second uses innovation outputs such as patents and new products, processes, and so on. For SMEs, collecting R&D data and patents is hard, and it is impossible to combine he anonymous BEEPS survey with other datasets t for firm-level studies. Among others Guisado-González et al. (2016), bocquet et al. (2017), and Attia et al. (2021) used dummy variables to define product innovation, process innovation, product and/or process innovation ( bocquet et al., 2017) and radical innovation (Guisado-González et al., 2016). Indeed, this measure is consistent with the OECD/ EUROSTAT (2005) innovation definition “Innovation encompasses the creation and adoption of new products, processes, marketing methods or organisational practices” (OECD/Eurostat, 2005). It is true that the concepts might overlap since, for instance, a green innovation (new energy measures developed by the company in our case) can be also new to the market. Yet, since we aim to investigate the characteristics that define this innovation considering it as green as well as radical innovation is correct.

Please note:  it is possible to build an innovation construct using the different innovation dummies. Yet, we will not be able to distinguish between the different characteristics of innovation.

We are grateful for your interesting remark. We have underlined it as one of our investigation’s limits. 

Also, the approach to medication effect by Baron and Kenny is outdated. Please consider the full mediation model using a conditional indirect effect analysis with bias-corrected bootstrap resampling by Hayes (2017)’s PROCESS procedure.

Thank you for your e have changed the estimation method as recommended (the empirical study is modified) please see the paper.

It’s disappointed that the study ignores the high correlations in table 2 without any explanations. It created a huge issue of multicollinearity, which makes the following analyses meaningless. Moreover, the correlation table didn’t show the correlations among the four DVs, which I am highly concerned about. Based on the descriptions of the first three DVs, I couldn’t find a clear difference. 

We are thankful for this important remark. We excluded the FEM_OWN variable from the control variables list. In addition, we add the innovation measures in the correlation matrix. Positive correlation between the different innovation measures is expected. We note that there is no correlation that exceed  0.5 between the DVs.

Conclusions:

It appears that the conclusion just summarized the results and didn’t contain any specific convinced contributions.

We modified the conclusion: we considered some implications and presented the limits of our research

“The informal sector reduces the formal companies' market share, which affects profitability, repayment abilities, and lending activity. Yet, it has a positive effect on SMEs' innovation. Hence, regulators should control the informal activity while encouraging formal companies to innovate due to other motivations. The main reason for informal sector development is the unemployment rate and the presence of an inadequately educated workforce that finds the informal economy as an alternative. Hence, programs and opportunities for the unemployed to get adequate training consistent with the markets' needs and which develop the workforce creativity should limit the informal competition while maintaining innovativeness. In addition, given the moderating effect of elaborating business strategies and the remarkable low average of SMEs that prepare formalized business strategies, it would be better for those firms to follow business financial literacy programs. Our study presents some limits. The binary consideration of the innovation measures and the credit constraints proxy cannot reflect the different levels and importance of the SMEs' innovative efforts. Therefore, future studies could examine whether the conditional indirect link of the informal competition affects innovation intensity. “  

Round 2

Reviewer 1 Report

Dear authors,

Thank you very much for incorporating my comments on the current version; from my viewpoint the paper is accepted in the current format.

Author Response

Dear Reviewer, 

We are really thankful for your efforts and for your considerable remarks that aim at improving our manuscript.

Reviewer comment: 

Thank you very much for incorporating my comments on the current version; from my viewpoint the paper is accepted in the current format.

In the new version, we highlighted the link between the special issue's theme and the paper. Besides, we undertake slight English revision.